# Conformational Dynamics of the RNA G-Quadruplex and its Effect on Translation Efficiency

**DOI:** 10.3390/molecules24081613

**Published:** 2019-04-24

**Authors:** Tamaki Endoh, Naoki Sugimoto

**Affiliations:** 1Frontier Institute for Biomolecular Engineering Research (FIBER), Konan University, 7-1-20 Minatojima-Minamimachi, Chuo-ku, Kobe 650-0047, Japan; t-endoh@konan-u.ac.jp; 2Graduate School of Frontiers of Innovative Research in Science and Technology (FIRST), Konan University, 7-1-20 Minatojima-Minamimachi, Chuo-ku, Kobe 650-0047, Japan

**Keywords:** conformational dynamics, co-transcriptional folding, co-translational refolding, metastable structure, G-quadruplex, translation suppression

## Abstract

During translation, intracellular mRNA folds co-transcriptionally and must refold following the passage of ribosome. The mRNAs can be entrapped in metastable structures during these folding events. In the present study, we evaluated the conformational dynamics of the kinetically favored, metastable, and hairpin-like structure, which disturbs the thermodynamically favored G-quadruplex structure, and its effect on co-transcriptional translation in prokaryotic cells. We found that nascent mRNA forms a metastable hairpin-like structure during co-transcriptional folding instead of the G-quadruplex structure. When the translation progressed co-transcriptionally before the metastable hairpin-like structure transition to the G-quadruplex, function of the G-quadruplex as a roadblock of the ribosome was sequestered. This suggested that kinetically formed RNA structures had a dominant effect on gene expression in prokaryotes. The results of this study indicate that it is critical to consider the conformational dynamics of RNA-folding to understand the contributions of the mRNA structures in controlling gene expression.

## 1. Introduction

An mRNA sequence not only determines the sequence of amino acids in protein but also controls gene expression by the formation of secondary and tertiary structures [1,2,3,4]. Thus, RNA structures present in the transcriptome of an organism are of interest since the structural elements formed by the RNA directly impact gene expression [5,6,7,8].

Owing to the directional nature of the transcription reaction, RNA structure elements are folded from 5′ to 3′ direction, a process referred to as co-transcriptional folding [9,10]. Co-transcriptional RNA folding restricts the landscape of its structure formation, which also engages the formation of metastable RNA structures [11]. These metastable structures are transient and show conformational transition to thermodynamically stable ones during the directional folding or post-transcriptional organization of the RNAs. The structures of coding regions of mRNAs are also reorganized co-translationally following passage of the ribosome, which incorporates and discharges single-stranded mRNA as the template strand. Thus, the intracellular RNA structures are transient and fluctuate dynamically over time as the processes involved in gene expressions occur.

G-quadruplexes are non-canonical structure elements formed by guanine-rich (G-rich) sequences in the DNA and RNA [12,13]. Since G-quadruplexes are stabilized under molecular crowding conditions, which mimics intracellular molecular environment [14,15,16], it is speculated that these structures have biological functions [17,18]. Previous studies have indicated that G-quadruplexes formed on the template strands of DNAs and RNAs become a roadblock of proteins that moves progressively on the template strands such as DNA polymerase, RNA polymerase and ribosome [19,20,21,22,23]. G-quadruplexes formed by mRNAs suppress progression of the ribosome along both non-coding (5′-untranslated) and coding (open reading frame, ORF) regions, and reduce their levels of protein expression [23,24,25,26,27]. G-quadruplex-mediated suppression of elongation during translation in the coding region also affects the ribosomal frameshift and nascent protein folding [28,29,30]. While multiple studies have implicated various functions of the G-quadruplex [17,31,32,33,34], a recent study suggested that the G-quadruplexes in eukaryotic cells are found in a globally unfolded state [35]. Additionally, an impairment in the translation and growth of bacteria caused by the G-quadruplex suggested an evolutionary depletion of G-quadruplex–forming sequences in prokaryotes [35]. However, we previously found five G-quadruplex–forming sequences in the ORF of the *E. coli* genes, including *glyQ*, which encodes the glycyl-tRNA synthetases and is fundamental for gene expression [23]. Since these findings are contradictory, to ascertain their accuracy, it would be necessary to discuss the dynamic behavior of RNA G-quadruplexes in cells [36]. The kinetics of formation and dissociation of the G-quadruplex are significantly slower than those of the canonical secondary structures [37,38,39,40]. The slow folding rate of the G-quadruplex is disadvantageous for its co-transcriptional folding and increases the possibility to form alternative metastable structures such as hairpins. When the translation reaction progresses before the metastable mRNA structure transitions to the stable G-quadruplex, its effect as a roadblock of the ribosome would be sequestered. Particularly, in prokaryotic systems, translation begins during transcription immediately after the ribosomal binding site is synthesized on the mRNA and discharged from the RNA polymerase. If metastable RNA structures are formed prior to formation of the G-quadruplex, a balance between the rates of transition from the metastable to the G-quadruplex structures and progress of the translation reaction may have a significant effect on gene expression. Thus, effects of the folding dynamics of G-quadruplexes on translation reactions should be elucidated to understand the characteristic processes involved in gene expression by co-transcriptional translation.

In the current study, we demonstrated that the metastable hairpin-like structure formed co-transcriptionally or co-translationally and the time lag between transcription and translation are the key factors that affect formation of the G-quadruplex, and thereby, suppression of translation both in vitro and in *E. coli* cells. The rate of transition from metastable to stable structures enables dynamic control of gene expression suggesting that a code for temporal gene expression is present in many mRNA structures. 

## 2. Results

### 2.1. Design of RNAs that Potentially Form Metastable Hairpin-Like and Stable G-Quadruplex Structures

It has been demonstrated that the G-rich sequence derived from the ORF of the *E. coli EutE* gene forms a stable G-quadruplex and suppresses the process of elongation during translation in mammalian cells [23]. The wild-type G-rich sequence contains two cytosine nucleobases at the 5′ end that flanking of the G-rich region and several cytosines in regions of the loops, which connect the guanine tracts involved in G-quartets formation. The sequence forms a metastable hairpin-like structure with several G-C base pairs, in which the thermodynamic stability (Δ*G*°) predicted by the Mfold program [41] is −11.3 kcal mol^−1^ (Figure 1b). The metastable hairpin-like structure transitions to the thermodynamically stable G-quadruplex in response to the potassium ion [42]. Thus, we expected a dynamic behavior of the G-rich sequence during its involvement in the processes of gene expression in the cells. Here, we designed RNA derivatives based on the wild-type G-rich sequence. Mutant A was designed to disrupt formation of the G-quadruplex. It has five mutations from guanine to adenine at positions involved in the formation of G-quartets. Mutant B and mutant C were designed to destabilize the metastable hairpin-like structure, while they maintained the numbers of the guanine tracts. Predicted hairpin-like structures in mutant B (with two mutations compared to the wild-type sequence) and mutant C (with four mutations) are significantly less stable than those of the wild-type sequence (Figure 1b). All sequences encode the same amino acids to enable evaluation of gene expression levels by analyzing the luminescence signal of the reporter protein, *Renilla* luciferase (Figure 1a).

Formation of G-quadruplexes by the RNA oligonucleotides (shown in Figure 1b) was evaluated by circular dichroism (CD) spectroscopy and RNase T1 digestion after refolding in buffers containing potassium at physiological concentrations (100 mM KCl). The CD spectra of the wild-type, mutant B and mutant C oligonucleotides were characterized by positive and negative peaks at 265 nm and 240 nm, respectively (Appendix A). These peaks are characteristics of parallel G-quadruplex structures. While the CD spectrum of mutant A had peaks at around 265 nm and 240 nm, a large negative peak at 210 nm suggested the formation of a hairpin-like structure partially forming an A-form RNA duplex (Figure 1b) [43]. Formation of G-quadruplexes by wild-type, mutant B, and mutant C, and hairpin-like structure by mutant A was also suggested by RNase T1 cleavage of the oligonucleotides (Appendix A). RNase T1 cleaved the wild-type, mutant B, and mutant C oligonucleotides at guanines G_15_ and G_25_, which are located at 3′ positions of the tracts of four guanines. Other guanines in the tracts were protected from the nuclease probably due to their involvement in the G-quartets. These results suggested that wild-type, mutant B, and mutant C oligonucleotides predominantly form parallel G-quadruplex structures containing three G-quartets in a buffer containing physiological concentrations of potassium (Appendix A). In contrast, significant RNase T1 cleavage was observed at G_13_, G_14_, and G_25_, which are locating loop and 3′ tail regions of the hairpin-like structure of mutant A (Figure 1b).

### 2.2. Formation of Metastable Hairpin-Like Structure during Transcription

Wild-type and mutant sequences were inserted into the 5′ region of the ORF encoding *Renilla* luciferase as the reporter gene (Figure 1a). DNA templates prepared using PCR were transcribed in vitro by T7 RNA polymerase in a buffer containing 100 mM potassium. After a transcription reaction at 37 °C for 30 min, TURBO DNase was added to the reaction mixture to degrade the DNA templates. Using agarose gel electrophoresis, it was confirmed that similar amounts of mRNA transcripts were obtained from all the templates (Figure 2a). *N*-methyl mesoporphyrin (NMM), a fluorescence indicator of the G-quadruplex structure [23,25,30], was added to the intact transcripts to evaluate formation of the G-quadruplex during transcription (Figure 2b). The fluorescence signals of NMM mixed with the mRNAs of mutant B and mutant C were significantly higher than those mixed with the control sample, which consisted of a solution for performing the transcription reaction without the DNA template, and with mutant A mRNA, which does not form a G-quadruplex. In contrast, the fluorescence signal of NMM mixed with the wild-type mRNA was almost equivalent to that of the mutant A mRNA. These results indicated that the wild-type mRNA does not form a G-quadruplex, whereas the mRNAs of mutant B and mutant C, which have the same guanine tracts as well as the wild-type mRNA but contain mutations, formed G-quadruplexes during transcription.

The NMM fluorescence signals were also evaluated with mRNAs that were denatured at 70 °C and refolded by cooling to 25 °C (Figure 2c); this treatment favors the formation of the structure that has maximum thermodynamic stability. Fluorescence of NMM mixed with the wild-type mRNA had increased significantly after refolding compared to that mixed with the intact wild-type mRNA without refolding (Figure 2b,c). This indicated that the G-quadruplex was more stable than the kinetically favored structure formed during transcription as an oligonucleotide of the wild-type G-rich sequence formed a G-quadruplex in a buffer containing 100 mM potassium (Appendix A). It is considered that the hairpin-like structure of wild type mRNA (as shown in Figure 1b) was formed as a kinetically favored metastable structure during transcription and had suppressed formation of the G-quadruplex. The fluorescence of NMM mixed with mutant B also increased after refolding of the mRNA. The small hairpin structure of mutant B is predicted to have a Δ*G*° of −2.3 kcal mol^−1^ (Figure 1b). The small increase of NMM fluorescence after refolding of mutant B mRNA suggests that some population of this mRNA formed the small hairpin structure co-transcriptionally. The fluorescence signals of NMM mixed with mRNAs of mutant A and mutant C were similar before and after the refolding, suggesting that the mRNA of mutant A does not adopt a G-quadruplex structure, whereas that of mutant C forms one irrespective of the folding conditions.

Time-course analyses of NMM fluorescence during in vitro transcription [44] were found to support co-transcriptional folding of metastable hairpin-like structures in the wild-type and mutant B mRNAs (Appendix A). Although the fluorescence signal of NMM increased with the reaction time of transcription for both mutant C and mutant B mRNAs, the rate of increase in signal in the mRNA of mutant B was slower. This indicated that the mRNA of mutant B first formed the metastable small hairpin structure co-transcriptionally and then transitioned to the more stable G-quadruplex. The Δ*G*° of the predicted hairpin-like secondary structure of the mutant C oligonucleotide was positive (Figure 1b); therefore, the mutant C mRNA had probably adopted the G-quadruplex structure during transcription without competition from any other metastable structure. In the case of wild-type mRNA, an increase of NMM fluorescence with the reaction time, which is slightly faster than that of mutant A mRNA, suggested that a small population of mRNAs formed the G-quadruplex during or after transcription. An increase in fluorescence with the mRNA of mutant A is possibly due to a non-specific interaction of NMM with the long mRNA [30]. 

### 2.3. Effects of G-Quadruplex Sequestering on In Vitro Translation 

Translation of mRNAs with and without refolding was carried out in vitro. After reactions at 37 °C for 30 min, the translation was terminated by addition of RNase A and puromycin; the nuclease degrades the transcript and the antibiotic immediately terminates the elongation process [45]. Protein expression levels in the *E. coli* S30 extract were determined based on the luminescence signal obtained after addition of the substrate for *Renilla* luciferase (Figure 3a). The relative intensities of luminescence were inversely correlated to those of the fluorescence observed when transcripts were incubated with NMM (Figure 2b,c). There was a large difference between the levels of protein expression observed in mRNA of the wild-type with and without refolding. When mRNAs were not refolded, the *Renilla* luciferase signal obtained from the wild-type was similar to that of the mutant A. When the mRNAs were refolded prior to translation, the level of signal from the wild-type was similar to those from mutant B and mutant C. These results indicate that translation was suppressed by the G-quadruplex, which is consistent with our previous reports [23,25]. Contrarily, even though the G-quadruplex is a thermodynamically stable structure, the translation reaction was found to progress smoothly and the level of protein expression had increased when the kinetically favored, metastable, secondary structure was sequestering the G-quadruplex (Figure 3b).

### 2.4. Effects of Co-Transcriptional Translation and Co-Translational mRNA Refolding on Gene Expression

To evaluate the effects of continuous co- and post-transcriptional translation systems on the reporter gene expression, the time courses of the protein expressions were evaluated in multiple turn-over translation reactions. Here, in vitro protein synthesis using recombinant elements (PURE) system (PUREfrex^®^; GeneFrontier) [46,47] was used to circumvent the obscurant effect of degradation of the DNA templates and mRNAs during the reaction. To mimic the co-transcriptional translation system in prokaryotes, DNA templates were mixed with T7 RNA polymerase and a translation solution of the PURE system. Protein expression levels in the coupled reaction were evaluated over a time course of 120 min (Figure 4a). At 15 min, the levels of protein expression were comparable among the sequence variants. These results suggested that translation began immediately after the mRNA was discharged from the RNA polymerase before formation of the G-quadruplex, which suppresses the translation. In contrast, after the 60 min reaction time point, the protein expression levels from the mRNA of mutant C were significantly reduced compared to those of the other sequence variants. It is expected that the mRNA of mutant C formed the G-quadruplex without the formation of any metastable secondary structure after the ribosome had passed through the G-rich sequence region as it efficiently formed the G-quadruplex during transcription (Appendix A). Thus, the numbers of mutant C mRNA, which formed the G-quadruplex, possibly accumulated with the reaction time and progressively suppressed post-transcriptional translation in the multiple turn-over system (Figure 4b).

Time course of protein expression levels after addition of refolded mRNAs were also evaluated (Figure 4c). After translation for 15 min, the levels of protein expression of wild-type, mutant B, and mutant C mRNAs were found to be significantly lesser than those of the mRNA of mutant A. We had expected these results because all mRNAs except the mutant A adopted a G-quadruplex structure after refolding and suppressed translation (Figure 3a). Signals of relative luminescence in mRNAs of the wild-type, mutant B, and mutant C at the 15-min time point were 46%, 37%, and 24%, respectively, compared to that of the mutant A mRNA. Despite significant differences in the levels of protein expression at 15 min, an increase in the rate of protein expression after 15-min time point for the wild-type mRNA was very similar to that of the mutant A mRNA. This result suggests that wild-type mRNA co-translationally refolded the metastable hairpin-like structure after the ribosome passed through the G-rich sequence region and was no longer able to suppress the translation (Figure 4d). The levels of relative protein expressions of the wild-type, mutant B, and mutant C mRNAs at the 120-min time point were 80%, 55%, and 29% compared to that of mutant A mRNA, respectively. The metastable small hairpin structure formed during co-translational folding of the mutant B mRNA transitioned to the G-quadruplex significantly faster than that formed by the wild-type mRNA. Thus, in the experimental condition without additional transcription from the DNA template, it was observed that the mRNA of mutant B caused moderate repression of translation. 

### 2.5. Protein Expression in E. coli Influenced by the Metastable mRNA Structure

Plasmid vectors which code the reporter gene constructs were transformed into the *E. coli* strain BL21(DE3). Expression of reporter proteins was induced by the addition of 100 μM isopropyl β-D-1-thiogalactopyranoside either in the absence or presence of 0.5 μM tetracycline. Tetracycline prevents binding of the aminoacyl-tRNA to its ribosomal acceptor site by interaction with the 30S ribosomal subunit [48]. After incubation for 60 min, the protein expression levels in *E. coli* were evaluated based on the luminescence signals from the cell lysates. The luminescence signals were normalized based on the optical density at 600 nm when the *E. coli* were lysed. In the absence of tetracycline, the levels of protein expression were very similar among all the mRNA variants (Figure 5a). Based on the results of the in vitro translation coupled with transcription at the 15-min time point, it was conjectured that the levels of protein expression were similar among the variants, because the mRNAs were translated before they formed the G-quadruplex (Figure 4a). Contrary to the normal translation conditions, in the presence of tetracycline, the level of protein expression from the mRNA of mutant C was significantly reduced compared to that of the other sequence variants (Figure 5b). Reduction in the activity of the ribosome would provide a time lag between the transcription of mRNA and initiation of the translation. Under these conditions, mutant C mRNA was expected to form the G-quadruplex structure without formation of the metastable secondary structure and suppressed the translation reaction in *E. coli*. Similar results were also observed in the presence of 2 μM chloramphenicol, which inhibits the elongation reaction during translation by binding to the 50S ribosomal subunit (Appendix A). 

## 3. Discussion

The rate of folding of RNA G-quadruplexes is generally slower than other simple secondary structures and, thus, would be influenced by their entrapment into kinetically favored metastable structures. Here, we designed mRNA derivatives based on G-rich sequences located in the ORF of the *E. coli EutE* mRNA to evaluate the effect of metastable hairpin-like structures with different stabilities (Figure 1b) that compete with the formation of thermodynamically stable G-quadruplexes during protein expression. To facilitate a direct comparison between the levels of protein expressions using luminescence signals, mutations were designed without changing the sequence of their amino acids (Figure 1a). The CD spectra and RNase T1 digestion revealed that all the designed sequences except mutant A adopted the thermodynamically favored G-quadruplex structure after refolding (Appendix A). In contrast, when NMM fluorescence was used as a probe for G-quadruplex formation, it was indicated that, if the G-rich sequence has a potential to form alternative secondary structures, kinetically favored metastable secondary structures are formed during transcription instead of the thermodynamically stable G-quadruplex; this suppressed its function as a roadblock of ribosomes (Figure 2 and Figure 3). In vitro multi turnover translation also indicated that the metastable secondary structures were co-translationally formed and consequently sequestered the G-quadruplex (Figure 4). 

In *E. coli*, mRNAs are quickly degraded by endogenous nucleases with an average half-life of several minutes [49,50]. It was considered that the protein expression in all the sequence variants were similar levels in *E. coli* (Figure 5a) because the mRNAs were translated and degraded before the sequences transitioned to the G-quadruplex (Figure 5c). Additionally, inhibition of ribosomal activity by antibiotics in *E. coli* facilitated the suppression of translation from mutant C mRNA (Figure 5b and Appendix A). It is considered that a decrease in the ribosome activity provided a window time to the mutant C mRNA to form the G-quadruplex structure that resulted in the translation suppression. These results suggested that in prokaryotic gene expression with a co-transcriptional translation system, the effect of RNA structure elements on the translation reaction was dominated by kinetically formed RNA structures. 

While G-quadruplex is a thermodynamically stable structure, the wild-type sequence derived from the ORF of the *E. coli EutE* gene did not reduce the level of protein expression due to formation of the kinetically favored metastable structure. Our data suggested that formation of a metastable hairpin-like structure enables *E. coli* to express the *EutE* gene, which encodes acetaldehyde dehydrogenase, to utilize ethanolamine as a carbon source [51]. All five G-quadruplex-forming sequences found previously in the ORF of the *E. coli* genes efficiently suppressed the ribosome progression under conditions in which the thermodynamically favored G-quadruplex was formed after refolding [23]. In each of these natural sequences, the mRNAs are predicted to form relatively stable hairpin-like structures that potentially sequester the G-quadruplex structure. These hairpin-like structures have Δ*G*° values ranging from −5.1 to −11.9 kcal mol^−1^ which are more stable than those of the mutant B mRNA (Appendix A). These metastable hairpin-like structures should form co-transcriptionally to enable efficient expression of the genes by disrupting formation of the thermodynamically stable G-quadruplex. This may be one of the reasons for these secondary structures to be evolutionarily conserved within the coding regions [52,53].

Co-transcriptional folding of RNA is one of the general mechanisms to functionalize both noncoding and coding transcripts in vivo [9]. The translation reaction in which a ribosome incorporates and discharges the mRNA as single strand also forces directional folding of the mRNA co-translationally. The directionality of RNA folding may lead to the formation of kinetically entrapped metastable structures that would, in turn, influence the resulting function of the RNA [54,55,56,57,58]. The dynamic features of intracellular RNAs are probably a reason why predictions of RNA structures based on thermodynamics have a less than perfect correlation with those characterized in vivo [6]. Evidences suggest that metastable RNA structures formed during directional folding transition rapidly to thermodynamically stable ones in vivo [59,60]. However, as demonstrated in the present study, if the metastable structures are sufficiently stable over a considerable timeframe from a biological standpoint, the metastable structures existing transiently will impact the biological reactions [11]. Particularly, in microorganisms, which have a relatively short lifecycle, the influence of metastable RNA structures would be relatively significant [54,57]. Even in eukaryotes, recent computational studies of RNA structures have suggested that the metastable RNA structures have an impact on gene modulations [61,62,63]. The impacts of the metastable RNA structure elements will vary depending on various kinetic factors such as the rates of transcription and translation, transition of the metastable structures, and translocation and degradation of the RNAs, respectively. Therefore, when the influence of metastable RNA structures on gene expressions is compared between eukaryotes and prokaryotes with different kinetic factors involved in the processes of gene expression, we will be able to provide important findings on the role of metastable RNA structures in biological systems.

## 4. Materials and Methods 

### 4.1. CD Spectrum Acquisition

RNA oligonucleotides purchased from Japan Bio Services Co., Ltd. (Saitama, Japan). were incubated in a buffer containing 50 mM Tris-HCl (pH 7.6), 5 mM magnesium acetate, 100 mM KCl, 2 mM spermidine, and 0.01% (*v*/*v*) Tween 20 at 70 °C for 5 min and cooled to 4 °C at a rate of 1 °C min^−1^. The CD spectra were collected on a JASCO (Tokyo, Japan) J-820 spectropolarimeter at 37 °C in cuvettes of 1.0 mm path length. The CD spectra shown are averages of three scans. The temperature of the cell holder was regulated by a JASCO PTC-348 temperature controller. 

### 4.2. Partial Digestion of RNA Oligonucleotide by RNase T1

RNA oligonucleotides were labelled with Alexa Fluor 546 C5-maleimide (Thermo Fisher Scientific, Waltham, MA, USA) using the 5′ EndTag Nucleic Acid Labelling System (Vector Laboratories, Burlingame, CA, USA). Labelled RNAs were purified using a denaturing polyacrylamide gel and precipitated with ethanol in the presence of 20 µg glycogen. Five pmol labelled RNAs in a transcription-buffer (T-buffer) containing 50 mM HEPES-KOH (pH 7.6), 5 mM magnesium acetate, 100 mM potassium glutamate, 2 mM spermidine and 0.01% (*v*/*v*) Tween 20 were incubated at 70 °C for 5 min and cooled to 37 °C at a rate of 1 °C min^−1^. The samples were incubated with 0.02 U of RNase T1 (Roche, Basel, Switzerland) at 37 °C for 10 min and electrophoresed on a 20% denaturing polyacrylamide gel at 70 °C. The fluorescence signal in the gel was imaged using a FLA-5100 fluorescence image scanner (Fuji Film, Tokyo, Japan) with 532 nm excitation and 575 nm emission.

### 4.3. Construction of Reporter Plasmids

Plasmid vectors encoding wild-type or mutant A sequences in the ORF of *Renilla* luciferase (pCMV-TnT-G-rich-RL) were previously constructed based on pCMV-TnT vector (Promega, Madison, WI, USA) [19]. The reporter gene constructs on the pCMV-TnT-G-rich-RL plasmids were digested by *EcoR*I and *Not*I, and appropriate DNA fragments were inserted into same restriction enzyme sites of pET21a (+) (Novagen, Madison, WI, USA). DNA fragments encoding G-rich sequences of mutant B and mutant C (Appendix A) were purchased from Hokkaido System Science Co., Ltd. (Hokkaido, Japan) and inserted into the *EcoR* I and *Sal* I sites to yield variants of the pET21-G-rich-RL plasmids.

### 4.4. Preparation of the DNA Template

DNA templates encoding the G-rich sequences followed by the *Renilla* luciferase ORF were amplified from the pET21-G-rich-RL plasmids by PCR using the T7 promoter and terminator primers. Amplified DNA fragments were purified using the QIAquick PCR Purification Kit (Qiagen, Hilden, Germany).

### 4.5. In Vitro Transcription

DNA templates (20 ng/μL) were mixed with T7 RNA polymerase (2 U/μL) in T-buffer containing 1 mM rNTPs and incubated at 37 °C for 30 min. TURBO DNase (Thermo Fisher Scientific) was subsequently added to the reaction mixture at a concentration of 0.048 U/μL and incubated at 37 °C for 30 min. In order to refold the mRNA, transcripts were incubated at 70 °C for 3 min followed by cooling from 50 °C to 25 °C at a rate of 1 °C min^−1^.

### 4.6. Fluorescence Analysis of NMM

Aliquots (5 μL) of mRNA transcripts with or without refolding were diluted into 50 μL of T-buffer containing 500 nM NMM and 0.01% (*v*/*v*) dimethyl sulfoxide. Fluorescence signal of NMM was measured using a microwell plate reader (Varioskan Flash, Thermo Fisher Scientific) at 400 nm excitation and 615 nm emission wavelengths after incubation at 37 °C for 30 min.

### 4.7. In Vitro Translation

Aliquots (1 μL) of mRNA transcripts with or without refolding were diluted into 4 μL reaction solutions of the *E. coli* S30 Extract System (Promega) and incubated at 37 °C for 30 min. For in vitro translation using PUREfrex (Gene Frontier, Chiba, Japan), aliquots (0.5 μL) of intact or refolded mRNA were diluted into reaction solutions of 2 μL and incubated at 37 °C. The translation reaction was terminated by a 10-fold dilution of the translated product in phosphate buffered saline (PBS) containing RNase A (20 μg/mL) and puromycin (20 μM) followed by incubation at 37 °C for 10 min. Levels of *Renilla* luciferase were determined by measuring the luminescence signal after addition of 5 μM coelenterazine (Promega) on the Varioskan Flash.

### 4.8. Evaluation of Protein Expression Levels in E. coli

Cells of the *E. coli* strain BL21(DE3) were transformed with the pET21-G-rich-RL plasmids. The cells were cultured over night at 37 °C in 2 × YT media containing 50 μg/mL ampicillin. The cultured cells were diluted 200 times into fresh 2 × YT media and incubated at 37 °C. When they reached an optical density of 0.5–1.0, the cells were transferred to the wells of a 96-well plate and diluted 2-fold with fresh 2× YT media containing final concentrations of 100 μM β-D-1-thiogalactopyranoside and indicated antibiotics. After incubation for 60 min at 37 °C, the optical density of *E. coli* was measured at 600 nm using a microwell plate reader (Infinite M200 Pro, Tecan, Mannedorf, Switzerland). Cells were lysed in 10 μL media using 50 μL of Passive Lysis Buffer (Promega) and the luminescence signal of 2 μL aliquot of the lysate was measured after addition of coelenterazine (5 μM) using Varioskan Flash.

## Figures and Tables

**Figure 1 molecules-24-01613-f001:**
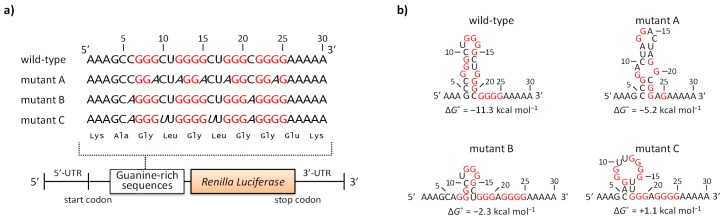
G-rich sequence variants designed to form metastable hairpin-like structures with different stabilities. (**a**) G-rich sequence elements and reporter mRNA constructs. Guanine nucleobases expected to be involved in the formation of the G-quadruplex structure are given in red. Nucleotides mutated from the wild-type sequence are indicated in italics. The amino acid sequence is indicated below the sequence of the nucleic acids and the reporter construct is shown schematically. (**b**) Secondary structures of the G-rich sequence variants predicted using the Mfold program. Thermodynamic stabilities (Δ*G*°) of the secondary structures predicted by the Mfold program are given.

**Figure 2 molecules-24-01613-f002:**
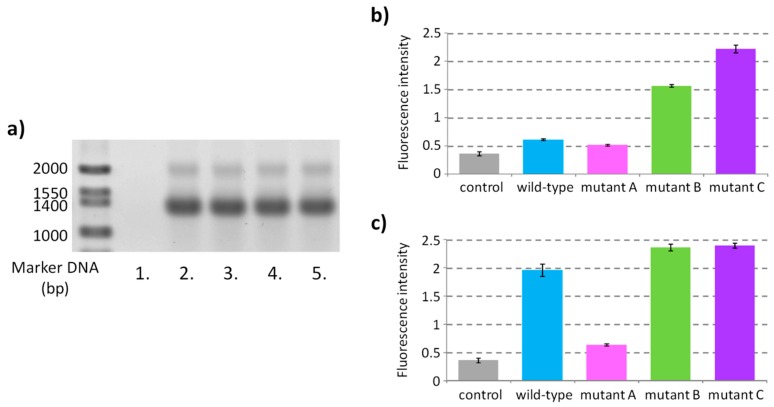
Co-transcriptional formation of the G-quadruplex depending on the stability of metastable hairpin-like structures. (**a**) mRNA transcripts of the reaction without DNA template (lane 1), with DNA template for wild-type (lane 2), mutant A (lane 3), mutant B (lane 4), and mutant C (lane 5) were electrophoresed on an agarose gel. mRNAs were stained with ethidium bromide and imaged at 532 nm excitation and 575 nm emission wavelengths. (**b**,**c**) Fluorescence intensities of NMM mixed with mRNA transcripts immediately after transcription (**b**) or after refolding by heating to 70 °C and then cooling to 25 °C (**c**). mRNA transcripts were diluted in a buffer containing 50 mM HEPES-KOH (pH 7.6), 5 mM magnesium acetate, 100 mM potassium glutamate, 2 mM spermidine, 0.01% Tween 20, 0.01% DMSO, and 500 nM NMM. NMM fluorescence was measured at 37 °C at 400 nm excitation and 615 nm emission wavelengths. Values are expressed as means ± S.D. of experiments performed in triplicates.

**Figure 3 molecules-24-01613-f003:**
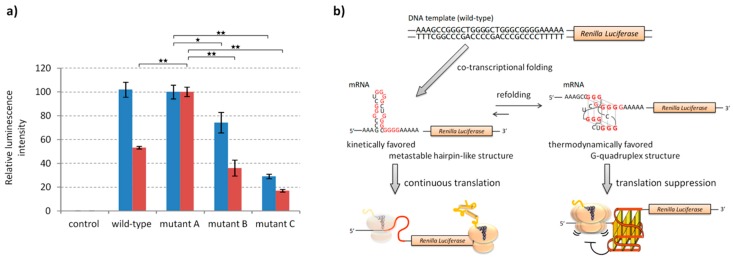
Protein expression levels from intact mRNA and after its refolding using the in vitro translation system of *E. coli* S30 extract. (**a**) Relative luminescence intensities from *Renilla* luciferase translated from mRNA transcripts with (red) and without (blue) refolding. Luminescence signals were measured after addition of coelenterazine (5 μM) to the translated products and were normalized to the signal obtained from mutant A mRNA. Values are expressed as mean ± S.D. of triplicate experiments. Asterisks indicate two-tailed *P*-values for Student’s *t*-test: * *P* < 0.05 and ** *P* < 0.01. (**b**) Effect of the mRNA structure elements formed during co-transcriptional folding or refolding of wild-type G-rich sequence on translation reaction.

**Figure 4 molecules-24-01613-f004:**
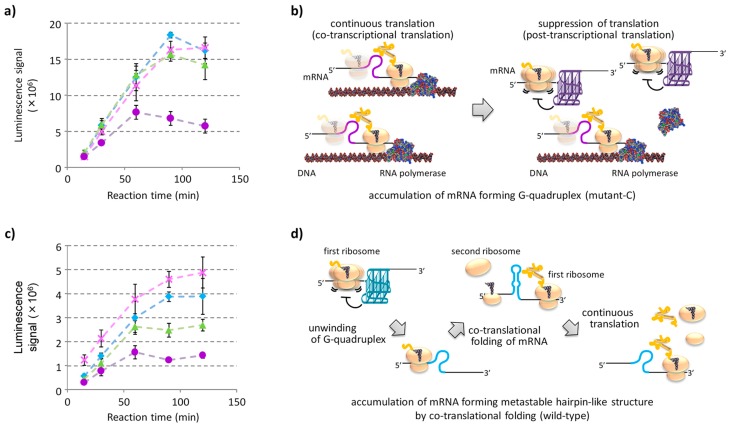
Time course of the levels of protein expression affected by co-transcriptional and co-translational RNA conformational dynamics within the hairpin-like and G-quadruplex structures. (**a**) Time course of co-transcriptional translation of reporter genes encoding wild-type (blue), mutant A (pink), mutant B (green), and mutant C (purple) sequences. DNA templates were mixed with the PURE system solution in the presence of T7 RNA polymerase and incubated at 37 °C. (**b**) Predicted suppression of translation caused by accumulation of the mutant C mRNA, which formed the G-quadruplex, with increasing reaction time. (**c**) Time course of translation of the refolded mRNAs with G-rich sequence of the wild-type (blue), mutant A (pink), mutant B (green), and mutant C (purple). Transcribed mRNAs refolded at 70 °C were mixed with PURE system solution and incubated at 37 °C. (**d**) Schematic of co-translational folding of metastable hairpin-like structure in wild-type mRNA that allows uninhibited translation by a subsequent ribosome. In (**a**) and (**c**), luminescence intensities of *Renilla* luciferase are expressed as mean ± S.D. of triplicate experiments.

**Figure 5 molecules-24-01613-f005:**
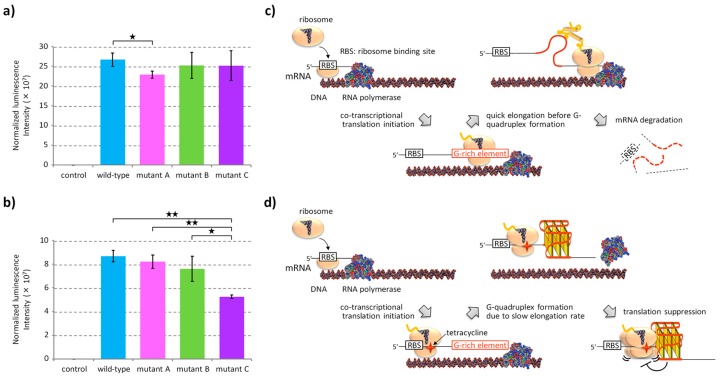
Gene expression in *E. coli* dominated by kinetically favored metastable mRNA structure. (**a**,**b**) Normalized luminescence intensities of the *E. coli* lysate cultured in the absence (**a**) or presence (**b**) of 0.5 μM tetracycline. Protein expression was induced by 100 μM β-D-1-thiogalactopyranoside in 2× YT medium containing 100 mM potassium glutamate for 1 h. Luminescence signals were normalized by adjusting to an optical density of 600 nm of *E. coli* cells. Values are expressed as mean ± S.D. of triplicated *E. coli* culturing wells. Asterisks indicate two-tailed *P*-values for the Student’s *t*-test: * *P* < 0.05 and ** *P* < 0.01. (**c**) Illustration of the co-transcriptional translation in usual culturing conditions of *E. coli*, in which the ribosome translates a region of G-rich elements before forming the G-quadruplex. (**d**) Illustration of co-transcriptional translation in the presence of tetracycline, in which the level of protein expression is dominated by kinetically favored mRNA structures. Only mRNA of mutant C, which forms the G-quadruplex by bypassing any metastable structure, reduces the level of protein expression due to the roadblock function of the G-quadruplex.

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
