# Peer review of "Conformational Dynamics of the RNA G-Quadruplex and its Effect on Translation Efficiency"

_molecules, 2019, doi:10.3390/molecules24081613_

Round 1
Reviewer 1 Report
The manuscript “Conformational dynamics of the RNA G-quadruplex and its effect on translation efficiency” by Tamaki Endoh and Naoki Sugimoto well describes the relevance of kinetically driven structural rearrangements on the regulation of transcription and translation. This is a critical issue in the frame of nucleic acids properties and, currently, it deserves deep investigation. The present work is well design and presented. Thus, it represents a helpful tool to move forward to the description of the multiple physiological functions of nucleic acids within the cells. I strongly support its publication in Molecules since it can provide large benefits for all Molecules readers.
Only a minor issue:
Page 7, line 262: the sentence is not clear, maybe rewrite as “..normalized based on the optical density at 600 nm….”
Author Response
Response to the reviewer
We really appreciate that the reviewer supported our manuscript to be published in Molecules.
It is our pleasure that the reviewer emphasized an importance of the physiological role of kinetically controlled nucleic acid structures that is the main focusing point in this study.
We revised the sentence (Page 7, line 262) according to the suggestion from this reviewer.
Reviewer 2 Report
The manuscript submitted by Endoh and Sugimoto is very elegant and the experiments have been nicely designed and performed. The concept according which RNA G4 structures could not form due to the formation of kinetically favoured hairpin-like structure is really interesting and I appreciated the sets of experiments performed to illustrate such an aspect.
I only have one concern and it is about the CD experiment. All sequences, wild type and mutants, share very similar bands in the experimental conditions used. Even mutant A, which is supposed to not form a quadruplex shares these bands. The authors report that mutant A shows a negative peak at 210 nm and they argue that, despite the similarity with the other mutants, only the presence of this band indicates the formation of an A-form RNA duplex. To be honest, this is not convincing. I would suggest to repeat the CD of the wild type and mutated sequences, recording the spectra at different temperatures. The melting curves obtained should provide significant differences between mutant A and the others. In the same way the authors did for the other experiments, It would also be interesting to evaluate the melting temperatures of the mRNAs (by CD) with and without a previous denaturation step to see the behaviour of the wild type sequence.
Once clarified the CD issue, I recommend publication of this manuscript in Molecules.
All the best
Author Response
Response to the Reviewer 2
We appreciate that the reviewer supported our manuscript to be published in Molecules.
We agree with the reviewer that the CD spectra in Supporting Figure 1a was not enough to show that the mutant A forms hairpin-like structure rather than the G-quadruplex. Thus, we performed RNase T1 cleavage assay in Supporting Figure 1b. The results of RNase T1 provided unambiguous results that is indicating formation of the hairpin-like structure in mutant A oligonucleotide. G13, G14, and G25 positions, which are locating loop and 3′ tail regions of the predicted hairpin-like structure of mutant A were clearly and significantly cleaved by RNase T1. Especially G13 and G14 positions of other oligonucleotides were protected due to their incorporation into G-quartets. Thus, considering together the CD spectrum and the RNase T1 cleavage assay, we think it is convincing that the wild-type, mutant B and mutant C formed G-quadruplex, and mutant A formed hairpin-like structure.
We revised our text in the revised manuscript to make our point clear that we evaluated the structure of the oligonucleotide variants by both CD spectra and RNase T1 cleavage.
Actually, we have previously analyzed CD spectra of similar sequences of wild-type and mutant A oligonucleotides in our previous study (Angew. Chem. Int. Ed. 2013, 52, 5522-5526). In the study we analyzed CD spectra of the wild-type and mutant A oligonucleotides under molecular crowding condition containing 40 wt% PEG200 at 37 degree C. There was clear difference in signal intensities between the wild-type and mutant A oligonucleotides. We attached the result of the CD spectra in attached file for this reviewer.
